# Vascular Smooth Muscle Cell Metabolic Disorders in the Occurrence and Development of Aortic Aneurysms and Dissections: Implications for Therapy

**DOI:** 10.3390/biomedicines13123072

**Published:** 2025-12-12

**Authors:** Yuqing Shi, Xianghuan Xie, Yang Sun, Yanghui Chen, Guangzhi Chen

**Affiliations:** Division of Cardiology, Department of Internal Medicine, Tongji Hospital, Tongji Medical College, Huazhong University of Science and Technology, 1095# Jiefang Ave., Wuhan 430030, China; yuqingshi2025@163.com (Y.S.); m202476596@hust.edu.cn (X.X.); sunyang.7@163.com (Y.S.)

**Keywords:** aortic aneurysms and dissections, vascular smooth muscle cells, metabolic dysregulation, phenotypic switching, programmed cell death, extracellular matrix remodeling, epigenetic modification

## Abstract

Aortic aneurysm and dissection (AAD) represent a life-threatening aortic disorder, for which current treatment strategies rely predominantly on surgical interventions, with limited pharmacological options targeting the underlying pathophysiology. Vascular smooth muscle cell (VSMC) dysfunction constitutes a central pathological mechanism in the development and progression of AAD. This review outlines the association between VSMCs and AAD, covering their physiological functions and pathological alterations, including phenotypic switching, cell death, and VSMC-mediated extracellular matrix remodeling. It further discusses the impact of epigenetic modifications on VSMC core functions. Additionally, this review addresses normal energy metabolism pathways of VSMCs and the mechanisms of metabolic reprogramming, as well as abnormalities in amino acid metabolism, lipid metabolism, and other metabolic pathways. Signaling mechanisms related to AMPK activation and mitochondrial function are also highlighted. Currently, AAD management is dominated by surgical interventions, while pharmacological therapy remains limited to symptomatic control. Looking ahead, future research should focus on VSMC metabolism-related mechanisms to develop early prevention strategies and novel targeted therapeutics, thereby improving the current treatment landscape for AAD.

## 1. Introduction

Aortic aneurysm and dissection (AAD) are life-threatening cardiovascular conditions due to a high risk of rupture [1]. Clinically, thoracic aortic aneurysm (TAA) and aortic dissection (AD) are often linked to hereditary connective tissue disorders (e.g., Marfan syndrome and Loeys–Dietz syndrome) or bicuspid aortic valve issues, whereas abdominal aortic aneurysm (AAA) is more strongly associated with traditional cardiovascular risk factors, particularly older age, male sex, smoking, and hypertension [2,3]. However, hereditary and sporadic AAD share the common histological feature of medial degeneration, which is associated with inflammation, apoptosis, vascular smooth muscle cell (VSMC) phenotype switching, and degradation of the extracellular matrix [4]. These processes lead to weakening of the aortic wall, aortic dilation, dissection, and ultimately rupture [5].

Current definitive management, while effective, carries substantial perioperative risks and demonstrates limited applicability in cases with smaller dimensions or complex anatomy [6]. Current management of AAD is largely determined by aortic size, growth rate, and the presence of symptoms or complications [7]. For selected patients, open surgical replacement of the diseased aorta or endovascular repair (TEVAR/EVAR) effectively prevents rupture and has become the standard of care once guideline-defined thresholds are met [8]. However, these procedures are not suitable for all patients. In patients with smaller aneurysms, medical therapy is limited to aggressive risk-factor modification and blood-pressure control, typically with β-blockers or renin–angiotensin system inhibitors [7].

Despite advances in surgical and endovascular interventions, pharmacological therapies that halt disease progression are lacking. This review, therefore, examines VSMC metabolic dysregulation as a central driver of AAD, integrating evidence on how VSMC phenotypic switching, programmed cell death, and extracellular matrix remodeling intersect with epigenetic regulation to destabilize the aortic wall. We then summarize abnormalities in glucose, amino acid, and lipid metabolism, and discuss key signaling and mitochondrial/TCA-cycle nodes that couple metabolic stress to aortic injury. Finally, we highlight VSMC metabolism–targeted strategies as promising avenues for early prevention and disease-modifying therapy in AAD (Figure 1).

This review uniquely centers on VSMC and organizes its narrative around metabolic dysregulation, integrating multiple pathways—including glucose metabolism, the BCAA–BCKDK–mTOR axis, dysregulated fatty acid and sphingolipid metabolism, and TCA cycle and mitochondrial dysfunction—into a cohesive “network map” within VSMCs. This integrated perspective remains relatively distinct among existing reviews. Moreover, unlike most VSMC-focused reviews that treat epigenetic regulation and metabolic abnormalities separately, this manuscript emphasizes the central concept of “metabolic intermediates serving as substrates or cofactors for epigenetic modifications.” It elaborates on the roles of AKG–TET2, lactate-mediated histone lactylation, citrate-driven histone acetylation, and the influence of the BCAA/mTOR axis on epigenetic regulation and VSMC fate determination, thereby linking metabolic products to DNA/histone modifications and ncRNA regulation as continuous pathways. This integrative content is largely absent in the current literature. Furthermore, the review systematically incorporates a substantial body of recent mechanistic studies from 2022–2025, significantly enriching the understanding of molecular regulatory networks in this field.

## 2. Association Between Vascular Smooth Muscle Cells and AAD

### 2.1. Physiological Functions of Vascular Smooth Muscle Cells in the Aorta

The VSMCs are the predominant cellular component of the middle layer of the vascular wall, the tunica media. Contractile VSMCs are essential for maintaining vascular tone [9]. Arterial elasticity stems from VSMCs’ contraction and elastic fiber recoil. The VSMCs regulate hemodynamics via contraction and withstand aortic mechanical stress due to their inherent stiffness [10]. Consequently, VSMC dysfunction is a central contributor to diverse vascular pathologies, including atherosclerosis, hypertension, intimal hyperplasia, and pulmonary hypertension [9]. Genomic and functional studies have established that signaling pathways such as TGF-β, IGF, VEGF, and PDGF are critically involved in regulating elastic fiber homeostasis, VSMC function, and structural remodeling of the aortic wall [11].

### 2.2. Pathological Changes of Vascular Smooth Muscle Cells in AAD

#### 2.2.1. Phenotypic Switching of VSMC in AAD

VSMCs exhibit high plasticity and can switch between contractile and synthetic phenotypes. Under physiological conditions, the contractile phenotype predominates, with its differentiation primarily governed by transcription factors such as myocardin (MYOCD) and serum response factor (SRF) [12].

Under pathological stimuli, VSMCs undergo transdifferentiation toward a synthetic state, losing contractile function while acquiring the ability to proliferate, migrate, and secrete inflammatory mediators and matrix metalloproteinases (MMPs). This shift drives extracellular matrix degradation and weakening of the aortic wall [13,14]. AAD, as well as various cardiovascular diseases such as atherosclerosis and hypertension, are all closely associated with the phenotypic switching of VSMCs [1].

The contractile program is primarily driven by the transcription factor SRF, which binds to CArG boxes and recruits co-activators like myocardin to induce contractile gene expression [15]. This transcriptional axis is reinforced by factors such as PTEN, which stabilizes SRF binding at contractile gene promoters [16]. Conversely, pathological stimuli induce factors like KLF4, which disrupt the SRF–myocardin complex and suppress the contractile program [15]. A key inducer of this phenotypic switch is PDGF-BB, which activates the ERK1/2-MAPK pathway to disrupt SRF–myocardin activity and promote a synthetic state [17]. Fine-tuning of this network also occurs at the post-transcriptional level, as illustrated by the ALDH2/miR-31-5p axis, where modulation of miR-31-5p levels ultimately regulates myocardin expression and phenotype stability [18,19]. The TGF-β signaling pathway acts as a key “rheostat” that regulates VSMC phenotype and aortic wall remodeling. Its canonical Smad-dependent pathway enhances the expression and activity of SRF, thereby maintaining the contractile phenotype and ECM integrity [15]. In contrast, the imbalance characterized by the inhibition of the canonical pathway and activation of non-canonical pathways drives VSMCs to switch toward an inflammatory/matrix-degrading phenotype [20]. Yet, the precise conditions under which TGF-β signaling transitions from protective to deleterious—and how this transition is influenced by metabolic status, mitochondrial function, and extracellular cues—remain incompletely understood and represent critical knowledge gaps.

Cytosolic DNA accumulation in AAD activates the cGAS-STING-TBK1-IRF3 axis, which orchestrates a dual pathogenic cascade: intrinsically inducing VSMC death (apoptosis/necroptosis) and extrinsically driving macrophage-mediated MMP-9 expression and ECM degradation [21]. In parallel, activated IRF3 orchestrates the epigenetic repression of contractile genes by recruiting EZH2 to deposit the repressive histone marker H3K27me3, thereby directly linking DNA sensing to VSMC phenotypic switching [22]. Notably, EZH2 also plays a context-dependent role in regulating autophagic cell death in VSMCs via the MEK-ERK pathway, indicating its multifaceted function in determining cell fate beyond epigenetic silencing [23]. Nevertheless, key questions remain unresolved, including how thresholds of cytosolic DNA burden are set in different AAD contexts and whether long-term pharmacologic STING inhibition can safely dampen pathogenic remodeling without impairing host defense.

Moreover, the TNF-OXPHOS-AP-1 axis drives the synthetic phenotype by inhibiting OXPHOS in VSMCs and activating the AP-1 transcription factor, while the inhibition of OXPHOS itself is a typical accompanying feature of glycolytic metabolic reprogramming [24].

#### 2.2.2. Programmed Cell Death in VSMC Associated with AAD Progression

Progressive VSMC loss is a hallmark of AAD. Ferroptosis is a novel iron-dependent form of regulated cell death, which is activated in AAD. The key regulators SLC7A11, FSP1, and GPX4 are downregulated, while the markers TFR, HMOX1 are upregulated. METTL3 accelerates VSMC ferroptosis via m^6^A-dependent degradation of SLC7A11/FSP1 mRNAs [25]. Related studies on Ferroptosis-1 (Fer-1) have further corroborated this notion. Ang II can induce iron-dependent lipid peroxidation and downregulation of GPX4 in VSMCs, while Fer-1 significantly inhibits VSMC ferroptosis and attenuates AAD formation by activating the SLC7A11/GPX4 axis [26]. The reactive products (elevated levels of iron, transferrin receptor, and 4-HNE) disrupt cellular function by adducting key biomolecules (e.g., proteins, lipids, nucleic acids) and impairing membrane integrity. This cytotoxic process is exacerbated by H3K9me-mediated epigenetic repression of antioxidant defenses (such as the System Xc-GPX4 axis), which promotes ferroptosis and AAD progression. BRD4770, a histone methyltransferase inhibitor, reduces H3K9me, restores antioxidant activity, thereby alleviating lipid peroxidation and ferroptosis. Furthermore, it mitigates the release of proinflammatory cytokines induced by ferroptosis, decreases inflammatory cell infiltration in the aortic wall, and ultimately prevents AAD development [27].

Autophagy plays a dual role in AAD. While basal autophagy protects VSMCs by alleviating ER stress, its genetic impairment (e.g., Atg5 deletion) exacerbates AAD [28]. Conversely, excessive autophagy can be pathological, as seen with reduced EZH2 expression promoting autophagic cell death via the MEK-ERK-ATG5/7 axis [23].

#### 2.2.3. Extracellular Matrix Remodeling Mediated by VSMCs in AAD

The ECM, a key structural and functional component of the vascular wall, confers distensibility and tensile strength to the aorta [2]. In AAD, dysfunctional VSMCs are the primary drivers of pathological ECM remodeling. They secrete excessive MMPs, leading to uncontrolled matrix degradation [29]. Similarly, proteases like LGMN exacerbate ECM breakdown and promote VSMC dysfunction, linking matrix degradation directly to altered VSMC fate [13]. Neutrophil-derived elastase further aggravates this process by degrading the ECM and driving VSMC phenotypic switching [30].

Taken together, these data suggest that VSMC phenotypic switching, programmed cell death, and ECM remodeling are not isolated events but components of a tightly interconnected response to hemodynamic and inflammatory stress. Imbalanced TGF-β signaling alters ECM composition and promotes an inflammatory, synthetic VSMC phenotype that secretes MMPs and cytokines; cytosolic DNA released from dying or stressed VSMCs activates the cGAS–STING axis in both VSMCs and macrophages, further amplifying inflammatory signaling, matrix degradation, and epigenetic repression of contractile genes; and mitochondrial dysfunction and oxidative stress lower the threshold for VSMC loss, thereby accelerating medial cell loss and wall weakening. A major unresolved question is how these pathways are hierarchically organized in vivo—for example, whether mitochondrial damage is the primary upstream trigger for cytosolic DNA accumulation and STING activation, or whether chronic inflammatory signaling secondarily impairs mitochondrial quality control. Addressing these issues will be essential to identifying which nodes in this network are most promising for therapeutic intervention.

## 3. Regulation of Epigenetic Modifications on the Core Functions of VSMCs

Epigenetic modifications are instrumental in regulating gene expression without causing permanent changes to the DNA sequence [31]. While genetic mutations contribute to AAD, epigenetic mechanisms are increasingly recognized as pivotal. These encompass DNA methylation, m6A RNA modification, histone modifications, and non-coding RNA actions [32].

### 3.1. TET2-Mediated DNA Demethylation Maintains the Contractile Program of VSMCs

The TET protein family (TET1/2/3) is are α-ketoglutarate-dependent dioxygenase that catalyzes active DNA demethylation by converting 5-methylcytosine to 5-hydroxymethylcytosine (5hmC) [31]. Among them, TET2 is the most abundantly expressed in VSMCs and is a key regulator of phenotypic transition. TET2 and 5hmC are enriched in contractile VSMCs but diminished in synthetic VSMCs. Accordingly, TET2 knockdown suppresses pro-contractile genes (e.g., myocardin, SRF) and upregulates synthetic markers (e.g., KLF4) [33].

Mechanistically, TET2-mediated promoter demethylation facilitates the binding of SRF and myocardin to CArG boxes, driving contractile gene expression. This is potentiated by a permissive histone modification landscape enriched with activating marks (e.g., H3K9ac, H3K4me2). Conversely, TET2 inhibition in synthetic VSMCs promotes promoter hypermethylation and loss of activating histone marks, though histone H3 lysine 4 Dimethylation (H3K4me2) often persists, suggesting a latent potential for contractile gene reactivation [34,35].

### 3.2. Histone Modification Programs Governing VSMC Identity

Nucleosomes, the fundamental units of chromatin, consist of 147 DNA base pairs wrapped around a histone octamer. The N-terminal tails of histones undergo diverse post-translational modifications—including acetylation, methylation, and phosphorylation—that critically regulate gene expression by modulating chromatin structure and transcription factor accessibility [36]. For example, histone acetylation generally promotes an open chromatin state, whereas deacetylation compacts chromatin [34,37]. Histone methylation effects are site-specific; H3K4me3 is activating, while H3K27me3 is repressive [38].

During VSMC differentiation, H3K4me2 enrichment at VSMC gene promoters (e.g., ACTA2, MYH11) establishes a stable epigenetic signature maintaining lineage identity [39]. At contractile gene CArG boxes, H3K4me2 recruits TET2 to sustain contractile expression. H3K4me2 loss induces promoter hypermethylation, impairing contractility and enhancing transdifferentiation potential. This modification also regulates VSMC migration via miR-145, collectively establishing the H3K4me2/TET2/miR-145 axis as a molecular memory system governing VSMC identity [35]. In contrast to H3K4me2, histone H3 lysine 9 dimethylation (H3K9me2) is a classic repressive histone mark. In models of vascular injury and atherosclerosis, the global level of H3K9me2 in VSMCs decreases, and H3K9me2 is significantly removed, especially at the promoters of inflammatory response genes such as MMPs and IL6. The demethylation at these sites exacerbates the induction of NF-κB target genes and inflammatory responses in the vascular wall, thereby promoting vascular remodeling [40].

In a parallel inflammatory pathway, IL-6/STAT3 signaling recruits the histone demethylase JMJD2B to erase repressive H3K9me3 at the RUNX2 promoter, driving osteogenic transition and vascular calcification [41].

### 3.3. MicroRNA-Mediated Epigenetic Regulation in AAD

Non-coding RNAs (ncRNAs) are categorized into housekeeping and regulatory types, with the latter including small ncRNAs like microRNAs (miRNAs)—key epigenetic regulators with over 2600 identified in humans [42].

A cohort study identified 12 differentially expressed miRNAs in AAD, including upregulated miR-15a and miR-659, and downregulated miR-1183 and miR-192 [43]. Functionally, miR-26b-3p inhibits ADAM17, promotes contractile markers, and reduces inflammation. However, in AAD, LXRα is upregulated and recruits UHRF1 to epigenetically silence miR-26b via promoter hypermethylation, thereby contributing to ECM degeneration, inflammation, and phenotypic switching [44].

Another miRNA, miR-3154, is upregulated in early AAD and correlates with aneurysm size. It targets TNS1, disrupting the TNS1-TLN1 interaction and leading to Sp1 phosphorylation and subsequent upregulation of MEOX1, a novel regulator of phenotypic switching, forming the miR-3154-TNS1-TLN1-Sp1-MEOX1 pathogenic axis [45].

In cellular senescence, p53 induces miR-1204, which targets MYLK to promote a senescence-associated secretory phenotype and contractile loss in VSMCs, fueling inflammation and dedifferentiation. miR-1204 also establishes a positive feedback loop to perpetuate senescence and exacerbate AAD [46].

### 3.4. Outstanding Knowledge Gaps in Epigenetic Control of VSMCs in AAD

Although the above studies firmly establish that TET2-mediated DNA demethylation, H3K4me2 “lineage memory” marks, and specific miRNAs are indispensable for maintaining the contractile VSMC phenotype, several questions remain. The causal hierarchy between DNA methylation, histone modifications, and ncRNA regulation in driving VSMC fate decisions during AAD is still unclear. The extent to which metabolic intermediates (e.g., α-ketoglutarate, citrate, lactate, BCAAs) dynamically tune epigenetic enzymes in human AAD remains speculative, despite compelling preclinical evidence. Addressing these gaps will require longitudinal sampling, single-cell multi-omics, and functional perturbation studies in both experimental models and human tissue.

## 4. VSMCs Heterogeneity and Cell–Cell Communication Revealed

Recent developments in single-cell technologies (scRNA-seq and scATAC-seq) have greatly refined our understanding of AAD at single-cell resolution [47]. In AAD, several recent scRNA-seq studies have mapped the cellular landscape of human ascending aorta and dissected aortic tissue and identified eight major cell types and 50 subpopulations, including multiple VSMC, endothelial, fibroblast, and immune cell subsets [48].

Within the AAD microenvironment, critical communication is established between GPNMB-high macrophages and VSMCs through the GPNMB/CD44 signaling axis. Induced by TNF-α, macrophages upregulate GPNMB, which is subsequently cleaved by ADAM10 to release soluble GPNMB (sGPNMB). These macrophages are localized—as shown by Seq-Scope and immunofluorescence—to areas of severe elastic fiber degradation in the medial layer, in proximity to VSMCs, facilitating sGPNMB paracrine signaling. sGPNMB binding to VSMC surface CD44 initiates pathways (e.g., G-actin/MRTF/SRF) that suppress contractile markers and promote pro-inflammatory cytokine/MMP secretion. This cascade drives VSMC phenotypic switching, destabilizes the vascular wall, and accelerates ECM degradation. However, CD44 knockdown only partially reverses these effects and does not prevent the upregulation of IL6, CXCL5, MMP9, and ADAM10, indicating the involvement of auxiliary signaling pathways [49]. In addition, the necrosis of VSMCs is not an isolated event. The dsDNA released by necrotic VSMCs is phagocytosed by macrophages. The intracellular dsDNA then activates the cGAS-STING pathway, which in turn mediates TBK1-dependent activation of IRF7. Activated IRF7 translocates into the nucleus to induce the production of type I interferons (IFNs). These IFNs bind to IFNAR to activate the JAK-STAT pathway, promoting the expression of interferon-stimulated genes (ISGs) and ultimately inducing the differentiation of monocytes into interferon-inducible monocytes/macrophages (IFNICs). Additionally, IRF7 regulates pyroptotic molecules and inflammatory pathways in IFNICs, exacerbating vascular damage [50].

Moreover, single-cell studies have also begun to dissect AAD-associated VSMC plasticity and its epigenetic regulation. By combining lineage-tracing models, scRNA-seq and scATAC-seq, Chakraborty et al. demonstrated that contractile VSMCs in murine AAD transition into an inflammatory VSMC state characterized by up-regulation of interferon-stimulated genes and chemokines; chromatin-accessibility analyses implicated a dsDNA–STING–TBK1–IRF3 signaling axis and specific enhancer reprogramming in driving this phenotypic switch, and pharmacologic inhibition of STING or TBK1 mitigated aneurysm formation and rupture [22].

Beyond macrophages, VSMC crosstalk with other cell types—such as via the CXCL12-CXCR4/ACKR3 axis—is critical for AAD progression. Synthetic VSMCs are known to express high levels of the chemokine CXCL12. In AAD, neutrophils and adventitial fibroblasts are identified as major sources of the receptors CXCR4 and ACKR3, respectively. This axis is thought to recruit neutrophils and induce fibroblast transdifferentiation, thereby exacerbating AAD. Additionally, downstream pathway prediction shows that the binding of CXCL12 to CXCR4 can also activate pathways such as JAK-STAT and PI3K-Akt, further regulating cell chemotaxis, differentiation, and apoptosis, and thus participating in the pathological mechanism of AAD [48].

What’s more, macrophages can secrete TNF-α, which binds to TNFR1 on the surface of VSMCs to directly promote VSMC apoptosis. Additionally, the TGF-βsecreted by macrophages can bind to TGF-β receptors on VSMCs, activating the TGF-β signaling pathway within VSMCs and thereby exacerbating medial layer degeneration. Furthermore, VSMCs are not merely passively affected; the fibroblast growth factor 1 (Fgf1) they secrete acts as a paracrine signal, targeting fibroblasts, macrophages, and endothelial cells. This disrupts the normal physiological states of these cells, further amplifying the pathological damage response in the aortic wall and forming a vicious cycle that drives the progression of AAD [51].

All in all, these approaches deconvolute the cellular composition of diseased aortae, uncover phenotypic heterogeneity and lineage transitions of VSMCs and other vascular cells, and delineate complex cell–cell communication networks that cannot be appreciated by bulk transcriptomic analyses [47] (Figure 2).

## 5. Metabolic Abnormalities of VSMC in AAD

### 5.1. Metabolic Reprogramming of Glucose Pathways in VSMCs and Its Role in AAD

Energy metabolism in VSMCs is central to their function, coupling mitochondrial oxidative phosphorylation (OXPHOS) with cytosolic glycolysis to sustain contractile activity [52]. Paradoxically, VSMCs predominantly rely on glycolysis (>90% of glucose use), a phenomenon known as the Warburg effect, even under normoxia [53]. This glycolytic phenotype, characterized by lactate accumulation and impaired OXPHOS, is accentuated in AAD and promotes VSMC proliferation by meeting the high bioenergetic demands of synthetic states [54,55]. Consequently, synthetic VSMCs exhibit enhanced glycolytic flux and elevated expression of glycolytic enzymes (e.g., PKM2, PDK4) [56]. The metabolic regulator TRAP1 promotes vascular senescence by driving aerobic glycolysis and lactate accumulation. This facilitates histone H4K12 lactylation, which activates senescence-associated secretory phenotype genes and accelerates VSMC senescence [57]. As a key glycolytic product, L-lactate serves as a central metabolite that bridges cellular metabolism with epigenetic regulation. It directly remodels chromatin architecture and gene expression by serving as a substrate for lysine lactylation modifications on both histone and non-histone proteins. In activated M1 macrophages, the Warburg effect drives enhanced aerobic glycolysis, marked by increased activity of enzymes including LDH and PFKFB3. This metabolic shift results in substantial L-lactate accumulation, which can be metabolized into acetyl-CoA via enzymes such as ACSS2. Subsequently, histone acetyltransferases (e.g., p300) catalyze histone lysine lactylation (e.g., at H3K18 and H4K12). These lactylation marks promote an open chromatin conformation and regulate gene transcription. For example, in late-stage M1 macrophages, enrichment of H3K18la at promoters of repair genes (e.g., ARG1) facilitates their phenotypic transition from pro-inflammatory to reparative states. Concurrently, lactylation of non-histone proteins, exemplified by PKM2 at lysine 62, enhances its enzymatic activity and reduces nuclear translocation. This indirectly inhibits glycolytic flux and downregulates pro-inflammatory genes, cooperating with histone modifications to fine-tune immune cell function [58].

Conversely, the long noncoding RNA H19 encodes a protein that maintains the contractile phenotype by inhibiting hnRNPA2B1-mediated alternative splicing of PKM, thereby suppressing the pro-glycolytic PKM2 isoform and aerobic glycolysis [59]. The E3 ubiquitin ligase March 2 catalyzes PKM2 polyubiquitination, promoting its tetramerization. This active form suppresses the glycolytic shift, reducing lactate and H3K18la levels. Consequently, March 2 deficiency enhances H3K18la at the p53 promoter, activating p53-mediated apoptosis and accelerating AAD [60].

Similarly, Poldip2 deficiency promotes a contractile phenotype by inhibiting TCA cycle enzymes, shifting metabolism toward glycolysis and the hexosamine pathway. This elevates UDP-GlcNAc, enhancing O-GlcNAcylation of PSMC1, which suppresses the ubiquitin-proteasome system (UPS). UPS inhibition stabilizes SRF, represses KLF4, and promotes MYOCD/MRTF-A-driven contractile gene transcription, thereby maintaining VSMC differentiation [61].

### 5.2. Abnormal Amino Acid Metabolism in VSMCs

Amino acid oxidation occurs primarily in the mitochondrial matrix, where deamination channels them into various metabolic pathways. Branched-chain amino acids (BCAAs) regulate cellular bioenergetics, mitochondrial biogenesis, and mTOR signaling, while inhibiting autophagy. Their metabolic dysregulation is implicated in cardiovascular pathogenesis [62]. Under physiological conditions, BCAAs are transaminated by BCAT2 to form BCKAs, which are subsequently decarboxylated by the BCKDH complex for entry into the TCA cycle and ATP production [63]. However, in AAD, upregulated BCKDK inhibits BCKDH via phosphorylation, leading to BCAA accumulation. This activates mTOR signaling, promoting VSMC phenotypic switching to a synthetic state while exacerbating mitochondrial ROS production and vascular inflammation [14].

Beyond BCAA metabolism, other dysregulated amino acid metabolism and protein function also contribute to AAD. In experimental AAD models, the protein abundance of protein arginine methyltransferase 1 (PRMT1), the transcription factor E2F7, and Sirtuin 6 (SIRT6) is markedly reduced, whereas the expression of senescence markers and inflammatory factors is elevated. PRMT1 catalyzes arginine methylation of E2F7, which enhances its protein stability by attenuating ubiquitin-mediated degradation. Stabilized E2F7 then promotes the transcriptional activation of SIRT6, ultimately suppressing VSMC senescence and protecting against aortic dissection [64]. Furthermore, quantitative proteomic profiling of plasma from healthy donors and Marfan syndrome patients revealed a significant increase in protein nitration in MFS. Consistently, murine models of MFS exhibit significant nitration of 24 aortic proteins, including ACTA2 and seven other cytoskeletal and extracellular matrix proteins. This nitration is potentially mediated by activation of the NO–sGC–PKG pathway [65].

Following vascular injury, the transcriptional regulator TEAD1 upregulation promotes VSMC proliferation via SLC1A5-mediated glutamine uptake and mTORC1 activation. Although TEADs are essential for VSMC differentiation during development, they are dispensable for maintaining ACTA2 expression in mature VSMCs [66].

Deficiency of the citrate transporter ANK leads to intracellular citrate accumulation, which in turn enhances the acetylation of specific histone residues (H3K23, H3K27, and H4K5) and drives the transcription of pro-inflammatory genes in vascular smooth muscle cells. This promotes pro-inflammatory phenotypic switch and exacerbates aneurysm formation. Independent of citrate transport, ANK also inhibits vascular calcification by facilitating ATP release and increasing extracellular PPi levels [67].

Alpha-ketoglutarate (AKG), an endogenous TCA intermediate, functions as a critical metabolic regulator with broad cellular regulatory functions. For example, the JMJC family of histone demethylases can be classified into AKG-dependent dioxygenases, using AKG as an indispensable cofactor. In addition, AKG serves as a vital cofactor for enzymes such as TET1, TET2, and TET3 [31]. Accordingly, TET2 overexpression has been shown to restore the expression of VSMC contractile genes. Similarly, AKG treatment upregulates both TET2 protein expression and its catalytic activity, thereby counteracting vascular calcification [33].

Together, these observations argue that mitochondrial dysfunction and altered TCA-cycle flux are not merely epiphenomena in damaged aortas, but upstream drivers of VSMC apoptosis, senescence, inflammatory activation, and altered cell–cell communication. Nuclear receptor-mediated repression of ACO2, excessive PC-driven anaplerosis, and dysregulated OGDH–succinate signaling illustrate how discrete metabolic lesions can be translated into structural remodeling and rupture risk. However, several issues remain unresolved: how early these mitochondrial defects arise relative to clinical aneurysm detection; whether distinct mitochondrial phenotypes characterize thoracic versus abdominal, hereditary versus sporadic, or early- versus late-stage AAD. Clarifying these aspects will be essential for rationally incorporating mitochondrial-targeted therapies into AAD management.

### 5.3. Lipid Metabolism Dysregulation and Its Impact on VSMC Function in AAD

Dysregulated fatty acid metabolism significantly impacts vascular homeostasis. ELOVL6 deficiency alters the fatty acid profile, increasing palmitic acid and decreasing oleic acid levels, which triggers ROS production and AMPK activation. This activates KLF4, which upregulates p53/p21, suppresses mTOR signaling, and represses contractile markers. Consequently, this ELOVL6-AMPK/KLF4 axis drives VSMC growth arrest and phenotypic switching [68].

Sphingolipids regulate key processes in aortic pathology. Metabolomic studies reveal reduced levels of the ganglioside GM3 and its biosynthetic enzyme ST3GAL5 in AAD patients. GM3 binds directly to the extracellular domain of transferrin receptor 1, blocking its interaction with holo-transferrin. This inhibits cellular iron uptake in VSMCs and consequently suppresses ferroptosis. However, in AAD, upregulated KLF9 represses ST3GAL5 transcription, diminishing GM3 biosynthesis and thereby exacerbating pathogenesis [69].

Furthermore, apolipoprotein C-III (apoC-III) is highly upregulated in both the plasma and aortic tissues of AAD patients. Functionally, apoC-III inhibits proliferation and migration and promotes apoptosis in VSMCs, without inducing significant phenotypic switching. These effects are consistent with the characteristic loss of medial VSMCs observed in AAD and contribute to its pathogenesis [70].

In addition to the aforementioned mechanisms, linoleic acid/oleic acid reduces the risk of AAD. Specifically, Ang II activates group V secreted phospholipase A2 derived from endothelial cells in mice. This enzyme mobilizes linoleic acid and oleic acid, which in turn lowers the risk of AAD by promoting the expression of lysyl oxidase and alleviating endoplasmic reticulum stress. In contrast, short-chain fatty acids upregulate the expression of L-selectin and promote the release of cytokine-induced neutrophil chemoattractant 2αβ. This enhances the migration of neutrophils to inflammatory sites, participates in the initiation of AAD-related inflammation, and thereby indirectly affects the occurrence and development of AAD [71].

### 5.4. Related Molecular Mechanisms and Signaling Pathways

#### 5.4.1. AMPK: A Central Metabolic Regulator

AMPK (AMP-activated protein kinase), as a highly conserved master regulator of metabolism, is activated by sensing the AMP: ATP and ADP: ATP ratios. Its activation mechanisms include nucleotide-dependent canonical regulation (AMP/ADP binding to the γ subunit to promote Thr172 phosphorylation, protect it from dephosphorylation, and allosteric activation by AMP) and non-canonical regulation. Once activated, it regulates metabolism by phosphorylating a variety of substrates. In glucose and lipid metabolism, it promotes glucose uptake, glycolysis, and fatty acid oxidation, while inhibiting glycogen synthesis, lipid synthesis, and gluconeogenesis. In protein metabolism, it reduces protein synthesis by inhibiting mTORC1. It can also promote autophagy and mitophagy, and realize long-term metabolic reprogramming by regulating transcription factor [72]. In addition to its regulatory effect on metabolism, activation of adenosine monophosphate-activated protein kinase α can alleviate AngII-induced inflammatory response in macrophages and inhibit phenotypic switching of VSMCs by suppressing the phosphorylation and nuclear translocation of NF-κBp65 and STAT3 [73].

#### 5.4.2. Mitochondrial Dysfunction and TCA Cycle Metabolites in AAD Pathogenesis

Core metabolic pathways—including glucose, lipid, and amino acid metabolism—are indispensable for both normal vascular function and pathogenesis. Mitochondria, acting as cellular powerhouses, centrally regulate these processes via the TCA cycle and OXPHOS. Tight regulation of mitochondrial function is therefore essential for metabolic homeostasis. Notably, single-cell RNA sequencing has revealed widespread mitochondrial dysfunction across multiple aortic cell types, establishing it as a hallmark of AAD [74]. Oxidative stress results from an imbalance between ROS production and endogenous antioxidant defenses. In pathological states like AAD, this balance is disrupted, favoring oxidative damage [75]. Oxidative stress triggers VSMC apoptosis and phenotypic modulation [75,76]. In AAD, VSMCs exhibit persistent mitochondrial dysfunction and heightened oxidative stress. Excessive ROS (particularly superoxide anion) generated via mitochondrial dysfunction promotes DNA damage and premature senescence by impairing the Nrf2/ARE antioxidant pathway, compromising manganese SOD2 activity, and dysregulating mitochondrial dynamics and mitophagy. These changes collectively drive AAD progression by activating MMPs, inducing proinflammatory gene expression, and promoting apoptosis [77,78]. Consistent with this, antioxidant treatment with vitamin E attenuates AAD expansion and reduces rupture risk [78].

Metabolites derived from the TCA cycle play pivotal roles in AAD pathogenesis and represent promising therapeutic targets. In both human and experimental AAD, the activity of mitochondrial aconitase-2 (ACO2), a key TCA cycle enzyme, is diminished [4]. Nuclear receptors are ligand-inducible transcription factors that regulate metabolism. Among these, nuclear receptor subfamily 1 group D member 1 (NR1D1) shows the most pronounced dysregulation in AAD tissues. NR1D1 transcriptionally represses ACO2 expression by recruiting the NCOR1–HDAC3 corepressor complex. This repression disrupts ACO2 function, leading to mitochondrial metabolic dysfunction characterized by TCA cycle impairment, reduced respiratory capacity, and mitochondrial DNA (mtDNA) damage. Consequently, these deficits promote VSMC apoptosis and accelerate AAD progression [74]. Separately, NCOR1 interacts with FOXO3, NFAT5, and ATF3 to modulate contractile gene expression, Rgs1, and MMP levels, thereby influencing VSMC phenotypic switching [79].

Notably, supplementation with AKG, a metabolite downstream of ACO2, rescues mitochondrial function and suppresses AAD formation. Mechanistically, AKG inhibits the PXDN–HOCl–ERK axis, thereby attenuating oxidative stress, macrophage infiltration, elastin degradation, and collagen remodeling, which collectively curb aneurysm expansion [80]. Furthermore, AKG ameliorates abnormal lipid metabolism in mice by activating the AMPK–PGC-1α–Nrf2 pathway, thereby improving mitochondrial function and redox homeostasis [81]. Dyslipidemia is a well-established risk factor for cardiovascular disease.

Additionally, non-targeted metabolomics has identified succinate as a biomarker and therapeutic target in AAD. Mechanistically, oxoglutarate dehydrogenase (OGDH), a key upstream TCA enzyme directly involved in succinate synthesis, is critical. OGDH suppression reduces macrophage inflammation but elevates vascular mitochondrial ROS, aggravating AAD through the p38α–CREB–OGDH pathway [82].

Pyruvate carboxylase (PC) is a key component of the TCA. The expression of PC is significantly upregulated in AAD tissues. Overexpression of PC induces mitochondrial metabolic disorders, leading to nuclear and mitochondrial DNA damage. The DNA released into the cytoplasm activates the transduction of the STING signaling pathway, thereby causing VSMC necrosis [83].

The Nrf2/ARE pathway is a key defense system for the organism to resist oxidative stress-induced damage. Under basal conditions, Nrf2 binds to Keap1 and is degraded via the ubiquitin-proteasome system. When stimulated by oxidative stress or electrophilic substances, Keap1 undergoes a conformational change to release Nrf2. Free Nrf2 enters the nucleus, binds to ARE, and initiates the expression of downstream antioxidant enzymes and detoxifying enzymes, thereby maintaining cellular redox balance. This pathway is also regulated by p62-mediated autophagic degradation of Keap1, protein kinases (with positive regulation by PI3K, ERK, etc., and negative regulation by p38-MAPK, etc.), miRNA inhibition, and crosstalk with pathways such as NF-κB and Notch. It exerts protective effects in cardiovascular diseases through antioxidant, anti-inflammatory, and other actions [84]. The natural polyphenol pterostilbene competitively binds to Keap1, releasing Nrf2 and enabling its translocation to the nucleus. This activates the expression of downstream antioxidant genes (such as HO-1 and NQO1), thereby reducing the production of ROS and mitochondrial damage, and inhibiting the release of mtDNA. The reduced release of mtDNA suppresses the activation of STING and its downstream TBK1/IRF3/NF-κB signaling pathway, which in turn alleviates VSMC inflammation, decreases the levels and activity of MMPs, and maintains the contractile phenotype of VSMCs [6].

Collectively, these observations indicate that mitochondrial integrity serves as a central hub that links vascular energy metabolism, oxidative stress, and multiple programmed cell death pathways in VSMCs, thereby exerting a direct and causal influence on AAD initiation and progression (Figure 3).

#### 5.4.3. GSDMD–CHOP–ODC1-Putrescine Axis in AAD

In VSMCs, Gasdermin D upregulates the transcription factor CHOP. CHOP binds to the promoter of ODC1 to promote its expression, thereby increasing putrescine synthesis. High concentrations of putrescine promote the phenotypic switch of VSMCs and enhance the inflammatory response in human aortic smooth muscle cells through the C/EBPβ and NF-κB pathways. Meanwhile, it may exacerbate AAD by promoting the degradation of the fibrillar ECM, but it does not trigger the senescence of VSMCs [85].

#### 5.4.4. HMGB2 and PPARγ Regulation of VSMC Function

Studies have found that HMGB2 expression is increased in AAD patient samples, mouse models, and human aortic smooth muscle cells, making it an effective target for AAD treatment and prevention. Ang II activates the Wnt/β-catenin-LEF1 pathway to upregulate HMGB2 expression. HMGB2 decreases GLUT1 expression and promotes GLUT4 translocation, and affects the proliferation of VSMC by regulating changes in intracellular glucose levels mediated by the PPAR-γ/PGC-1α pathway [86]. What’s more, PPAR transcription factors regulate genes that encode proteins controlling metabolic homeostasis and function in multiple organs, including the liver, skeletal muscle, vascular wall, and heart [87]. Activation of PPARγ holds therapeutic potential for TGF-β1-associated pathologies, including pulmonary arterial hypertension and Marfan syndrome. In VSMCs, PPARγ is a key link between the TGFβ1 pathway and the anti-proliferative BMP2 pathway. Activated by BMP2, PPARγ inhibits both the canonical Smad3/4 pathway and the non-canonical STAT3–FOXO1 pathway induced by TGF-β1, through direct interaction with Smad3 and STAT3. Conversely, TGF-β1 downregulates PPARγ expression via induction of miR-130a and miR-301b. Additionally, PPARγ regulates the glycolytic enzyme platelet-type phosphofructokinase through miR-331-5p, thereby suppressing TGF-β1-induced mitochondrial activation and VSMC proliferation [88]. NPR-C deficiency activates the ERK1/2 pathway, reduces PPARγ expression and activity, suppresses hydroxyacyl–CoA dehydrogenase trifunctional multienzyme complex subunit beta expression, and disrupts mitochondrial homeostasis—particularly fatty acid β-oxidation. These changes collectively promote VSMC apoptosis, extracellular matrix degradation, and inflammatory responses. Notably, the NPR-C agonist C-ANP4-23 and spermidine attenuate AAD development by modulating these pathways [89].

## 6. Conclusions and Future Perspectives

The Stanford aortic dissection classification depicted the differences between type A and B aortic dissection. For type A dissections, surgical intervention involving excision of the intimal tear followed by graft replacement remains the gold-standard, life-saving treatment [90]. Type B dissections are managed with thoracic endovascular aortic repair (TEVAR) or medical therapy (e.g., α-blockers and β-blockers) [7]. However, medically managed type B dissections frequently progress to aneurysms, which are associated with elevated risks of aortic adverse events and mortality [91].

The Crawford classification for aortic aneurysm describes the extent of the aortic repair and has been adopted as the main classification system [92]. The size of an aneurysm is a universally recognized factor for predicting rupture, and treatment decisions need to balance the risk of rupture against the risk of surgery [93]. Endovascular AAA repair and open repair were the treatment options for AAAs [8].

A rich repertoire of druggable nodes within VSMC metabolic and stress-response pathways is outlined in this review: the KEAP1–Nrf2–STING axis, cGAS–STING–IRF3–EZH2 signaling, AMPK, NR1D1–ACO2–TCA cycle regulation, PC–cGAS–STING-mediated mitochondrial stress, and BCKDK–mTOR-driven BCAA accumulation. They concurrently regulate oxidative stress, inflammation, VSMC survival, phenotypic modulation, and extracellular matrix remodeling. Proof-of-concept studies utilizing agents including pterostilbene (a Nrf2 activator/KEAP1 inhibitor), AKG, ferroptosis inhibitors (e.g., BRD4770), and AMPK activators have demonstrated the feasibility of targeting these pathways in vivo; however, systematic evaluations of dose–response relationships, long-term safety, and reversibility in large animal models are required before clinical translation.

Pharmacological modulators are already available or in advanced development for several pathways discussed herein, including AMPK activators, PPARγ agonists, mTOR inhibitors, Nrf2 activators, and STING inhibitors/agonists. Rational repositioning of these agents may be achieved through alignment with specific VSMC metabolic phenotypes. For instance, Nrf2 activators or KEAP1 inhibitors could be investigated in patients exhibiting strong oxidative and inflammatory signatures to disrupt the KEAP1–Nrf2–STING feed-forward loop; PPARγ agonists may be utilized in cases where TGF-β1/BMP2–PPARγ–metabolism crosstalk predominates; and BCKDK inhibition might be assessed in genetically or metabolically defined patient subsets demonstrating significant BCAA accumulation and mTOR hyperactivation. Similarly, therapeutic strategies targeting ferroptosis, mitochondrial TCA flux could prove beneficial in carefully selected patient subgroups.

Successful clinical translation will also depend on biomarkers to identify patients most likely to benefit and to monitor on-target effects. Potential biomarkers include circulating and tissue-based measures such as succinate, BCAAs and their catabolites, specific lipid species (e.g., GM3), apoC-III, metabolite-associated epigenetic modifications (e.g., histone lactylation/acetylation), and disease-relevant miRNAs (e.g., miR-26b-3p, miR-3154, miR-1204), which could be employed as companion diagnostics for patient stratification and guidance of intervention timing and intensity. Integration of these biomarkers with imaging parameters (e.g., aortic diameter) would enable early-phase trials, thereby reducing dependence on hard clinical outcomes such as rupture or surgery.

However, AAD exhibits substantial heterogeneity, including distinctions between thoracic and abdominal locations, hereditary versus sporadic forms, and early-stage versus advanced lesions. This heterogeneity necessitates the integration of pathway-specific interventions with the current standard of care, thereby moving beyond a “one-size-fits-all” therapeutic strategy for AAD.

The VSMC metabolic, mitochondrial, and epigenetic targets discussed in this work collectively establish a conceptual framework for developing disease-modifying therapies that complement existing surgical and endovascular interventions. Future advancements will require integrated multidisciplinary efforts to translate these mechanistic insights into safe, effective, and personalized therapeutic strategies for AAD patients.

## Figures and Tables

**Figure 1 biomedicines-13-03072-f001:**
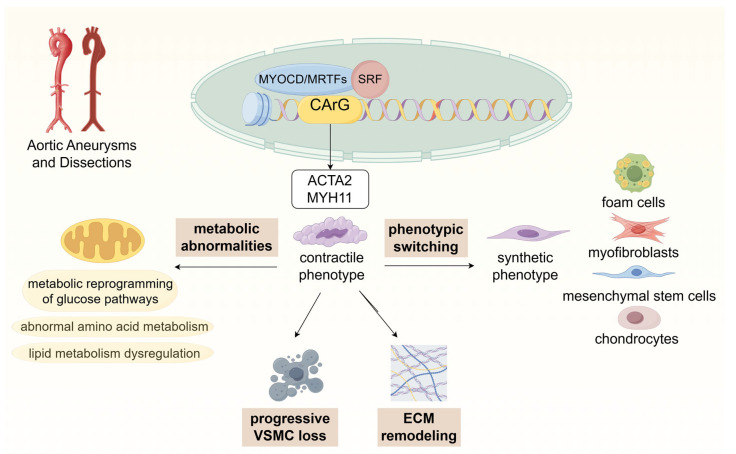
Schematic representation of key mechanisms contributing to VSMC dysfunction in AAD. VSMCs undergo metabolic reprogramming and phenotypic switching from a contractile to a synthetic state, driven by dysregulation in glucose, amino acid, and lipid metabolism. These changes promote ECM remodeling, progressive VSMC loss, and inflammation, ultimately leading to aortic wall weakening and AAD progression.

**Figure 2 biomedicines-13-03072-f002:**
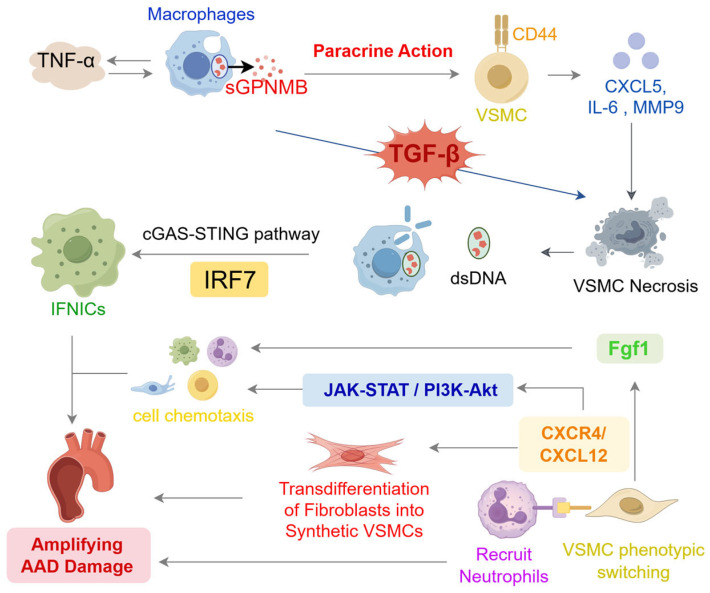
Schematic diagram of core cellular interactions and key signaling pathways during the pathological progression of AAD. TNF-α in the AAD microenvironment induces macrophages to secrete sGPNMB, which acts on CD44 on the surface of VSMCs in a paracrine manner, promoting their phenotypic switching and the release of factors such as IL-6 and MMP9. The dsDNA released by necrotic VSMCs activates the cGAS-STING-IRF7 pathway in macrophages to generate IFNICs. CXCL12 secreted by VSMCs can recruit neutrophils and induce transdifferentiation of fibroblasts, while Fgf1 amplifies pathological damage. Multiple pathways collectively drive the progression of AAD.

**Figure 3 biomedicines-13-03072-f003:**
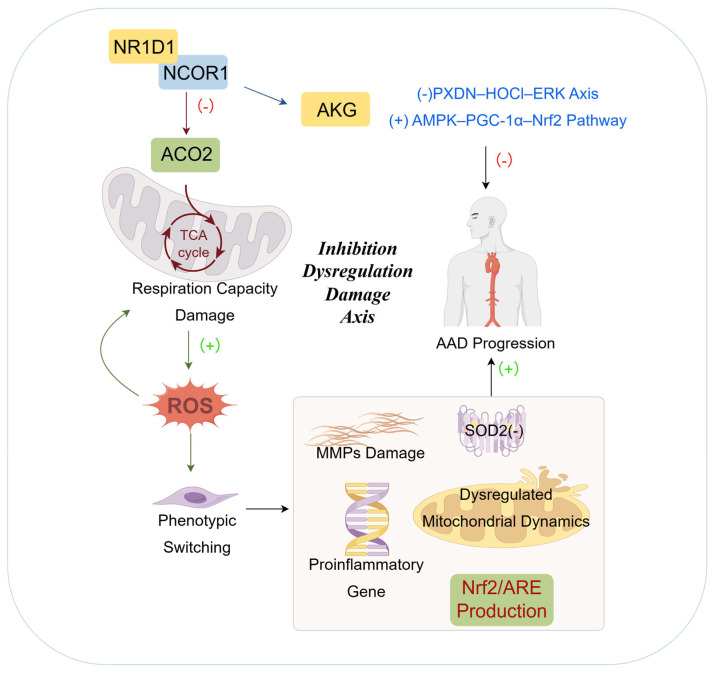
The core regulatory network of metabolic dysregulation and mitochondrial dysfunction in AAD. In AAD, the NR1D1 inhibits ACO2 (a key TCA cycle enzyme) by recruiting NCOR1, disrupting the mitochondrial TCA cycle. This impairs respiratory capacity and accumulates ROS, thereby driving phenotypic switching of VSMCs. This dysregulation is further associated with SOD2 inhibition, dysregulated mitochondrial dynamics, MMP damage, and proinflammatory gene activation, exacerbating aortic injury. AKG can rescue this dysregulation by modulating relevant pathways, thus suppressing AAD progression.

## Data Availability

No new data were created or analyzed in this study. Data sharing is not applicable to this article.

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
