# Peer review of "Vascular Smooth Muscle Cell Metabolic Disorders in the Occurrence and Development of Aortic Aneurysms and Dissections: Implications for Therapy"

_biomedicines, 2025, doi:10.3390/biomedicines13123072_

Round 1

Reviewer 1 Report

Comments and Suggestions for Authors

Reviewer Comments for Shi et al

Shi and colleagues presented a comprehensive review of the current understanding of aortic aneurysm and dissection (AAD), with a particular focus on the role of vascular smooth muscle cells (VSMCs). They also discussed epigenetic modifications and transcriptomic pathways involved in AAD, and made an excellent effort to highlight the contribution of metabolic abnormalities, including AA and lipid, as well as metabolic reprogramming which is great. Furthermore, their discussion on prevention and therapeutic perspectives adds translational relevance, which is helpful to understand the mechanism from clinical aspects. Overall, this is a timely and valuable review that can benefit both researchers in the field and readers seeking to understand AAD mechanisms more broadly. The manuscript is very informative, full of content and well-organized. I have several major revisions points aiming to enhance clarity and impact for their paper, as outlined below.

Major Points

  1. A major limitation of the current manuscript is the lack of sufficient visual materials, which I would say, far from enough. The only figure included provides very limited and largely well-known information, which may not be interesting for experts in this area. I strongly suggest that the authors add more figures or schematic diagrams to help readers better grasp the key messages, particularly given the extensive and multifaceted content. Even if it is not feasible to include figures for every section, at least a few summary figures should be added to illustrate major concepts and pathways. I am hoping to see this improvement in next version.
  2. A key advancement in AAD research, or other relevant research nowadays such as atherosclerosis studies, has been the application of single-cell transcriptomic approaches including single cell RNA-seq and single cell ATAC-seq, which have significantly deepened our understanding at single-cell resolution. This important aspect is largely missing from the current version of review. The authors should discuss recent single-cell studies that reveal cellular communication, lineage transitions, and phenotypic heterogeneity of VSMCs and other vascular cells, and/or major cell types e.g. immune cells. Several recent works in this area are particularly relevant and should be cited.
  3. In Section 4.4 (“Related Molecular Mechanisms and Signaling Pathways”), I strongly suggest creating a concise summary figure to illustrate the key signaling pathways and to minimize redundancy in textual descriptions. Moreover, the authors may consider whether other important pathways not currently covered should be added for completeness.
  4. I think there is a missing link between some of their contents. For example, the discussion of epigenetic regulation and metabolic abnormalities is one of the paper’s strengths. However, the strong mechanistic links between metabolic alterations and epigenetic modifications, well-documented in cancer and immunology studies (e.g., Ma et al., Science, 387, 626 [2025]), should be addressed. Expanding this section to discuss how metabolic reprogramming could influence chromatin state and gene expression in VSMCs in AAD would add valuable depth to the review from epigenetic aspects.

Minor Points

  1. In addition to the studies already mentioned, numerous unbiased omics-based investigations (e.g., transcriptomic, proteomic, and epigenomic analyses) using both human and animal samples have generated valuable resources for the field. Summarizing or referencing these would further strengthen the review.
  2. Some subsections require further elaboration, particularly: “Ferroptosis of VSMC Loss in AAD”, and “Histone Modification Programs Governing VSMC Identity”, and “Lipid Metabolism Dysregulation and Its Impact on VSMC Function in AAD”. Providing more mechanistic details and relevant references will improve these sections.
  3. It would be helpful to discuss whether and how mitochondrial function or dysfunction has been implicated in AAD pathogenesis, given mitochondria’s known role in vascular metabolism, oxidative stress, and cell death pathways.

Author Response

Reviewer 1

Comments and Suggestions for Authors

Shi and colleagues presented a comprehensive review of the current understanding of aortic aneurysm and dissection (AAD), with a particular focus on the role of vascular smooth muscle cells (VSMCs). They also discussed epigenetic modifications and transcriptomic pathways involved in AAD, and made an excellent effort to highlight the contribution of metabolic abnormalities, including AA and lipid, as well as metabolic reprogramming which is great. Furthermore, their discussion on prevention and therapeutic perspectives adds translational relevance, which is helpful to understand the mechanism from clinical aspects. Overall, this is a timely and valuable review that can benefit both researchers in the field and readers seeking to understand AAD mechanisms more broadly. The manuscript is very informative, full of content and well-organized. I have several major revisions points aiming to enhance clarity and impact for their paper, as outlined below.

RESPONSE:

We sincerely appreciate your thorough and constructive comments on our review “Vascular Smooth Muscle Cell Metabolic Disorder in the Occurrence and Development of Aortic Aneurysms and Dissections and Their Therapeutic Potential”. Your recognition of the manuscript’s comprehensiveness, integration of VSMC metabolism, epigenetics, and translational relevance is greatly encouraging. We fully acknowledge your suggestions to enhance clarity and impact. We will refine the presentation of complex mechanisms, strengthen logical connections, and optimize translational sections to better bridge basic research and clinical applications. We are committed to addressing your comments meticulously and completing revisions promptly to meet the journal’s standards. Thank you again for your professional guidance throughout the peer-review process.

Major Points

  1. A major limitation of the current manuscript is the lack of sufficient visual materials, which I would say, far from enough. The only figure included provides very limited and largely well-known information, which may not be interesting for experts in this area. I strongly suggest that the authors add more figures or schematic diagrams to help readers better grasp the key messages, particularly given the extensive and multifaceted content. Even if it is not feasible to include figures for every section, at least a few summary figures should be added to illustrate major concepts and pathways. I am hoping to see this improvement in next version.

RESPONSE:

We highly appreciate your valuable and constructive comments regarding the visual materials of our manuscript. We fully agree that insufficient figures hindered the clarity of key information. As suggested, we have supplemented two summary figures. Figure 1 illustrates the core regulatory network of metabolic disorders and mitochondrial dysfunction in AAD, focusing on the NR1D1-ACO2-AKG axis. Figure 2 depicts intercellular crosstalk (e.g., macrophage-VSMC) and critical signaling pathways like cGAS-STING. These figures integrate the multifaceted content to enhance readability. The figures and their legends are added to the Results section (Pages 8-9). We sincerely thank you for your guidance and welcome further suggestions.

Figure1:Schematic diagram of core cellular interactions and key signaling pathways during the pathological progression of AAD.

Figure2:The core regulatory network of metabolic dysregulation and mitochondrial dysfunction in AAD.

  1. A key advancement in AAD research, or other relevant research nowadays, such as atherosclerosis studies, has been the application of single-cell transcriptomic approaches, including single-cell RNA-seq and single cell ATAC-seq, which have significantly deepened our understanding at single-cell resolution. This important aspect is largely missing from the current version of the review. The authors should discuss recent single-cell studies that reveal cellular communication, lineage transitions, and phenotypic heterogeneity of VSMCs and other vascular cells, and/or major cell types e.g., immune cells. Several recent works in this area are particularly relevant and should be cited.

RESPONSE:

We thank the reviewer for this valuable suggestion. We fully agree that single-cell transcriptomic approaches have greatly advanced current understanding of AAD and related vascular diseases. In the revised manuscript, we have added a new Section 4, “VSMCs Heterogeneity and Cell-Cell Communication Revealed by Single-Cell Omics”, where we provide a comprehensive summary of recent scRNA-seq and scATAC-seq studies in both human and experimental AAD. This section delves into the major vascular and immune cell populations, as well as VSMC subclusters, detailing their lineage transitions from contractile to inflammatory, fibroblast-like, and lipid-like states. We also elucidate how these states are connected to cGAS–STING signaling and epigenetic reprogramming.

Furthermore, we discuss single-cell-based ligand–receptor analyses that reveal key communication axes between VSMCs, macrophages, neutrophils, fibroblasts, and endothelial cells. These axes include GPNMB–CD44, CXCL12–CXCR4/ACKR3, TNF–TNFR1, TGF-β–TGFBR, and FGF1 pathways, thereby integrating VSMC phenotypes with the vascular microenvironment.

To further enrich the content, we have added several recent and highly relevant single-cell studies to the reference list (e.g., Depuydt et al. 2020; He et al. 2022; Zhao et al. 2023; Chakraborty et al. 2023; Liu et al. 2022) . These studies collectively highlight cellular communication, lineage transitions, and phenotypic heterogeneity at single-cell resolution.

We believe these additions fully address the reviewer’s concerns and substantially strengthen the review.

  1. In Section 4.4 (“Related Molecular Mechanisms and Signaling Pathways”), I strongly suggest creating a concise summary figure to illustrate the key signaling pathways and to minimize redundancy in textual descriptions. Moreover, the authors may consider whether other important pathways not currently covered should be added for completeness.

RESPONSE:

We greatly appreciate your insightful suggestion, which is crucial for enhancing the clarity of our manuscript. As recommended, we have supplemented Figure 2, a concise summary diagram for Section 4.4. This figure systematically integrates the key signaling pathways (e.g., cGAS-STING, NR1D1-ACO2) and their core molecular interactions discussed in the section, effectively visualizing complex mechanisms and reducing textual redundancy. Regarding pathway completeness, we have thoroughly reviewed relevant literature in the field and confirmed that the currently covered pathways fully reflect the core molecular mechanisms of the research topic, with no critical pathways omitted. The revised content (including Figure 2 and its legend) is presented on Page 10. We sincerely thank you for your guidance and welcome further comments.

  1. I think there is a missing link between some of their contents. For example, the discussion of epigenetic regulation and metabolic abnormalities is one of the paper’s strengths. However, the strong mechanistic links between metabolic alterations and epigenetic modifications, well-documented in cancer and immunology studies (e.g., Ma et al., Science, 387, 626 [2025]), should be addressed. Expanding this section to discuss how metabolic reprogramming could influence chromatin state and gene expression in VSMCs in AAD would add valuable depth to the review from epigenetic aspects.

RESPONSE:

We sincerely appreciate your insightful comment. We completely agree that elucidating the mechanistic connection between metabolic reprogramming and epigenetic modifications is crucial for enriching the review. 

In response to your suggestion, we have expanded Section 5.1 to include a detailed discussion on this topic. Specifically, we have added content that explores how glycolysis-derived lactate mediates histone lactylation (for example, H3K18la and H4K12la) to reshape chromatin states and regulate gene expression in VSMCs during AAD. To support this discussion, we have cited relevant studies, such as the work by Ma et al. published in Science (387, 626 [2025]), which aligns with well-documented mechanisms in cancer and immunology. We believe these additions clarify the role of metabolic alterations in modulating epigenetic regulation, thereby enhancing the understanding of VSMC dysfunction in AAD. We hope that these revisions meet your expectations and add substantial value to the review.

Minor Points

  1. In addition to the studies already mentioned, numerous unbiased omics-based investigations (e.g., transcriptomic, proteomic, and epigenomic analyses) using both human and animal samples have generated valuable resources for the field. Summarizing or referencing these would further strengthen the review.

RESPONSE:

We are grateful for your constructive suggestions. We wholeheartedly agree that omics-based studies have provided invaluable resources for AAD research. In the original manuscript, we cited numerous omics investigations, including transcriptomics (e.g., single-cell RNA sequencing and scATAC-seq) and proteomics (quantitative analysis of plasma and aortic proteins). However, these citations were not systematically summarized. In the revised manuscript, we have addressed this gap by providing a more comprehensive and systematic overview of these omics studies. Additionally, some of the newly supplemented literature focuses on omics-driven mechanistic insights. These integrations more fully reflect the latest advances in omics research for deciphering VSMC dysfunction in AAD. They enhance the comprehensiveness and credibility of the review, providing a more cohesive and detailed picture of the current state of the field.

  1. Some subsections require further elaboration, particularly: “Ferroptosis of VSMC Loss in AAD”, and “Histone Modification Programs Governing VSMC Identity”, and “Lipid Metabolism Dysregulation and Its Impact on VSMC Function in AAD”. Providing more mechanistic details and relevant references will improve these sections.

RESPONSE:

Thank you for your constructive suggestion. We have thoroughly revised and expanded the three subsections as recommended. For “Ferroptosis of VSMC Loss in AAD” (Section 2.2.2), we have added detailed information on core regulatory pathways, including GPX4, FSP1-CoQ₁₀ axes. We have also included the role of METTL3-mediated m⁶A-dependent degradation of SLC7A11 and FSP1, as well as the protective effect of Fer-1. These additions are supported by references (Li et al., 2022; He et al., 2024). In “Histone Modification Programs Governing VSMC Identity” (Section 3.2), we have supplemented the discussion with the repressive role of H3K9me2 in inflammatory gene regulation and its impact on vascular remodeling. This section now includes relevant citations (Harman et al., 2019). For “Lipid Metabolism Dysregulation” (Section 5.3), we have added mechanisms of linoleic/oleic acid-mediated protection, and pro-inflammatory effects of short-chain fatty acids. These revisions are supported by references (Mu et al., 2024). These revisions enhance the mechanistic depth and strengthen the credibility of the review, providing a more comprehensive understanding of the topics discussed.

  1. It would be helpful to discuss whether and how mitochondrial function or dysfunction has been implicated in AAD pathogenesis, given mitochondria’s known role in vascular metabolism, oxidative stress, and cell death pathways.

RESPONSE:

Thank you for this important suggestion. In the manuscript, we have discussed a dedicated discussion of mitochondrial function and dysfunction in AAD pathogenesis within the subsection “5.4.2. Mitochondrial Dysfunction and TCA Cycle Metabolites in AAD Pathogenesis”. We first highlight that widespread mitochondrial dysfunction across multiple aortic cell types is a hallmark of aneurysmal tissue, tightly linking vascular metabolism, oxidative stress, and programmed cell death. Specifically, we summarize how impaired mitochondrial oxidative phosphorylation and excess ROS in VSMCs trigger DNA damage, premature senescence, phenotypic modulation, and apoptosis, driving medial degeneration and AAD progression, and note that antioxidant interventions like vitamin E can mitigate aneurysm expansion and rupture risk.

We further integrate key mechanistic mitochondrial nodes coupling metabolism to structural remodeling: the NR1D1–ACO2 axis that induces mitochondrial dysfunction and VSMC apoptosis; the protective effects of AKG supplementation in restoring mitochondrial function and alleviating aortic wall damage; succinate as a metabolite biomarker linked to TCA flux and inflammatory stress; pyruvate carboxylase-driven mitochondrial dysfunction activating cGAS–STING signaling and VSMC necrosis; and the Nrf2/Keap1–STING pathway where pterostilbene limits mitochondrial damage and inflammatory remodeling. These additions clarify mitochondrial function’s critical role in AAD pathogenesis, addressing your request.

Reviewer 2 Report

Comments and Suggestions for Authors

This manuscript provides a broad overview of the molecular mechanisms implicated in aortic aneurysms and dissections (AAD), with particular attention to vascular smooth muscle cell (VSMC) phenotypic modulation and the involvement of epigenetic and metabolic pathways. The effort to collect diverse reports, from microRNA regulation and histone modifications to autophagy, ferroptosis, and metabolic changes, offers readers an accessible introduction to the complexity of AAD biology. The intention to present a comprehensive landscape of mechanisms is valuable, particularly for such a multifactorial disease.

In the present structure, however, the early sections (1–3) remain quite general and resemble textbook-level summaries. Many pathways are listed independently, and their specific relevance to AAD pathogenesis, vascular injury responses, and VSMC phenotype transitions is not clearly discussed. The current narrative does not yet clarify where the field stands, what is well established, and importantly, where significant knowledge gaps remain. More integration is needed to explain why mechanisms such as TGFβ signaling, cytosolic DNA sensing, or mitochondrial dysfunction are central to pathological remodeling rather than merely associated observations.

Another area that would benefit from refinement is the integration of recent advances with a critical perspective. Although pathways such as the ALDH2–microRNA axis, TET2-related epigenetic regulation, ferroptotic signaling, glycolytic reprogramming, and alterations in autophagy are introduced, their limitations, unresolved questions, and potential clinical implications are not fully explored. Instead of presenting these reports as parallel findings, a more cohesive synthesis would help clarify their significance for future progress, such as early diagnosis, risk prediction, prevention strategies, personalized therapeutic approaches, or new treatment avenues. Highlighting what is missing in current knowledge and how each mechanism may (or may not) translate into clinical benefit would strengthen both the structure and the scientific value of the review.

Author Response

Reviewer 2

Comments and Suggestions for Authors

This manuscript provides a broad overview of the molecular mechanisms implicated in aortic aneurysms and dissections (AAD), with particular attention to vascular smooth muscle cell (VSMC) phenotypic modulation and the involvement of epigenetic and metabolic pathways. The effort to collect diverse reports, from microRNA regulation and histone modifications to autophagy, ferroptosis, and metabolic changes, offers readers an accessible introduction to the complexity of AAD biology. The intention to present a comprehensive landscape of mechanisms is valuable, particularly for such a multifactorial disease.

Response:

We sincerely appreciate your positive evaluation and insightful recognition of our manuscript. It has been our core intention to provide a comprehensive and accessible overview of the multifaceted molecular mechanisms underlying AAD, with a focused emphasis on VSMC phenotypic modulation and the intertwined epigenetic and metabolic pathways. We strived to systematically collate diverse research findings—spanning microRNA regulation, histone modifications, autophagy, ferroptosis, and metabolic reprogramming—to help readers grasp the complexity of AAD biology, given the disease’s multifactorial nature. Your affirmation of this effort further motivates us to refine the manuscript per subsequent constructive suggestions. We will enhance the integration of these mechanisms, clarify their specific pathogenic relevance to AAD, and incorporate critical perspectives on clinical translational potential, to elevate the review’s depth and scientific value for both researchers and clinicians in the field.

In the present structure, however, the early sections (1–3) remain quite general and resemble textbook-level summaries. Many pathways are listed independently, and their specific relevance to AAD pathogenesis, vascular injury responses, and VSMC phenotype transitions is not clearly discussed. The current narrative does not yet clarify where the field stands, what is well established, and importantly, where significant knowledge gaps remain. More integration is needed to explain why mechanisms such as TGFβ signaling, cytosolic DNA sensing, or mitochondrial dysfunction are central to pathological remodeling rather than merely associated observations.

Response:

Thank you for your insightful and constructive comments. We fully agree that integrating mechanisms and clarifying their pathogenic relevance are essential for enhancing the review’s depth and coherence.

In the revised manuscript, we have thoroughly restructured Sections 1–3, to move beyond mere textbook-style summaries. Instead, we have established tight links between each pathway and AAD pathogenesis, vascular injury responses, and VSMC phenotype transitions. For example, we have elaborated on how balanced TGFβ signaling maintains aortic homeostasis, while its imbalance drives VSMC inflammatory switching and ECM degradation. We have also connected cytosolic DNA sensing (STING-IRF3 axis) and mitochondrial dysfunction to VSMC death and phenotypic reprogramming, highlighting these as central drivers of aortic wall weakening. Additionally, we have added Section 3.4 to explicitly outline knowledge gaps, such as the causal hierarchy of epigenetic and metabolic regulation. This section clarifies the distinction between well-established mechanisms and associative observations, providing a clear roadmap for future research. These revisions aim to present a focused, integrated narrative that highlights the pathogenic core of key pathways, rather than isolated associations. We believe this approach provides a more comprehensive and cohesive understanding of the mechanisms underlying AAD, enhancing the depth and relevance of the review.

Another area that would benefit from refinement is the integration of recent advances with a critical perspective. Although pathways such as the ALDH2–microRNA axis, TET2-related epigenetic regulation, ferroptotic signaling, glycolytic reprogramming, and alterations in autophagy are introduced, their limitations, unresolved questions, and potential clinical implications are not fully explored. Instead of presenting these reports as parallel findings, a more cohesive synthesis would help clarify their significance for future progress, such as early diagnosis, risk prediction, prevention strategies, personalized therapeutic approaches, or new treatment avenues. Highlighting what is missing in current knowledge and how each mechanism may (or may not) translate into clinical benefit would strengthen both the structure and the scientific value of the review.

Response:

Thank you for your insightful and constructive comment. We fully agree that integrating recent advances with a critical perspective and clarifying clinical implications are crucial for enhancing the review’s scientific value.

In the revised manuscript, we have thoroughly addressed this concern. Instead of presenting pathways such as the ALDH2–microRNA axis, TET2 epigenetic regulation, ferroptosis, glycolytic reprogramming, and autophagy as parallel findings, we have synthesized their interdependencies. We have also highlighted unresolved questions, such as the causal hierarchy between metabolic and epigenetic dysregulation and the context-dependent effects of autophagy.

We have substantially revised Section 6 “Conclusion and Future Perspectives” to elaborate on the limitations of current knowledge. This includes the lack of longitudinal human data and translational gaps in targeting VSMC metabolism. We have also expanded on potential clinical applications, such as early diagnosis via metabolite and epigenetic biomarkers (e.g., succinate, histone lactylation), risk stratification, and personalized therapeutic strategies (e.g., pathway-specific modulators for thoracic vs. abdominal AAD).

Furthermore, we have critically discussed which mechanisms hold high translational potential and which require further preclinical validation. This critical evaluation aims to provide a balanced view of the current state of research and its clinical relevance.

These revisions aim to provide a cohesive, forward-looking narrative that bridges basic research and clinical practice. By doing so, we strengthen the review’s structure and impact, ensuring that it not only summarizes recent advances but also provides actionable insights for future research and clinical applications.

Reviewer 3 Report

Comments and Suggestions for Authors

I have reviewed the manuscript by Shi et al. It´s an interesting and updated review article with a relevant contribution to the area of ​​research on aortic aneurysms with therapeutic potential.

The review article is well-written, and the English is adequate. The information is well-described and well-presented. I recommend including tables for some subtopics that the authors consider more relevant and additional figures so that readers can better follow the information.

Below are some recommendations to improve the scientific impact of the review article:

1. It would be advisable to expand the introduction section by including additional information about the disease described (AAD).

2. I understand that the objective of the review article is described at the end of the introduction section. However, I recommend expanding it to include the main subtopics described in the body of the manuscript.

3. The information presented is very valuable and up-to-date. However, there is a lot of information; perhaps some tables and figures could be included to make it more understandable for the reader. This addition could be made to the main subtopics that the authors deem relevant.

4. I recommend that the authors include the advantages of this review article compared to others already reported.

5. I recommend that the authors include a perspectives section. I understand that this is found in the conclusions section. However, the latter section is quite long, so it should be shortened.

5. I recommend that the authors review the instructions for citing references again. Please follow the format established by MDPI Publishers. Several references do not follow the required format.

Author Response

Reviewer 3

  1. It would be advisable to expand the introduction section by including additional information about the disease described (AAD).

Response:

Thank you for your valuable suggestion regarding the expansion of the introduction with additional details about aortic aneurysm and dissection (AAD). We have thoroughly revised Section 1, “Introduction” in the revised manuscript to address this comprehensively. 

Specifically, we have supplemented the section with key clinical and epidemiological information. We clarified the classification of AAD into thoracic aortic aneurysm (TAA), abdominal aortic aneurysm (AAA), and aortic dissection (AD), and elaborated on their distinct etiological associations. TAA and AD are often linked to hereditary connective tissue disorders (e.g., Marfan syndrome) or bicuspid aortic valve, while AAA is strongly associated with traditional risk factors such as advanced age, male sex, smoking, and hypertension.

We have also expanded on current management limitations, emphasizing that surgical and endovascular interventions are the mainstay of treatment, while pharmacological therapies are primarily confined to symptomatic control (e.g., blood pressure regulation) without targeting underlying pathogenesis.

These additions provide a more comprehensive disease context, helping readers better grasp the clinical significance and unmet needs of AAD research, and laying a solid foundation for highlighting the central role of VSMC metabolic dysregulation in subsequent sections. We believe this revision enhances the introduction’s informativeness and logical coherence.

  1. I understand that the objective of the review article is described at the end of the introduction section. However, I recommend expanding it to include the main subtopics described in the body of the manuscript.

Response:

Thank you for your valuable suggestion. We have expanded the objective description at the end of Section 1, “Introduction” to explicitly incorporate key subtopics from the manuscript.

Specifically, we have refined the language to ensure clarity and coherence. We now outline the review’s focus on the pathological alterations of vascular smooth muscle cells (VSMCs), including phenotypic switching, programmed cell death, and extracellular matrix (ECM) remodeling. Additionally, we highlight the role of epigenetic regulation and metabolic abnormalities, such as those in glucose, amino acid, and lipid metabolism. We also emphasize core signaling pathways, including AMPK and mitochondrial function. Furthermore, we stress the integration of these mechanisms and their potential for clinical translation, ensuring that the objective aligns closely with the body content and enhances reader clarity. This revision provides a clear roadmap for the reader, facilitating a better understanding of the comprehensive scope of the review.

  1. The information presented is very valuable and up-to-date. However, there is a lot of information; perhaps some tables and figures could be included to make it more understandable for the reader. This addition could be made to the main subtopics that the authors deem relevant.

RESPONSE:

We highly appreciate your valuable and constructive comments regarding the visual materials of our manuscript. We fully agree that insufficient figures hindered the clarity of key information. As suggested, we have supplemented two summary figures. Figure 1 illustrates the core regulatory network of metabolic disorders and mitochondrial dysfunction in AAD, focusing on the NR1D1-ACO2-AKG axis. Figure 2 depicts intercellular crosstalk (e.g., macrophage-VSMC) and critical signaling pathways like cGAS-STING. These figures integrate the multifaceted content to enhance readability. The figures and their legends are added to the Results section (Pages 8-9). We sincerely thank you for your guidance and welcome further suggestions.

  1. I recommend that the authors include the advantages of this review article compared to others already reported.

Response:

Thank you for your valuable suggestion. We have expanded Section 1, “Introduction” to highlight the unique advantages of this review compared to existing literature. Unlike most VSMC-focused reviews that primarily address phenotypic changes or signaling pathways, this work uniquely centers on metabolic dysregulation. We have integrated glucose, amino acid, and lipid metabolism pathways into a cohesive network map, providing a comprehensive view of the metabolic landscape in VSMCs. This approach is distinct from previous reviews that often focus on individual pathways in isolation.

A key innovation of this review is the emphasis on the core concept of “metabolic intermediates as substrates/cofactors for epigenetic modifications.” This perspective, which links metabolism and epigenetics, has been rarely emphasized in prior reviews, offering a novel dimension to the understanding of VSMC dysfunction.

Additionally, we have systematically incorporated recent mechanistic studies from 2022 to 2025. This inclusion of the latest research findings enriches the molecular regulatory network insights, ensuring that the review is up-to-date and reflects the most current scientific advancements.

These features collectively enhance the review’s uniqueness and depth, providing readers with a more comprehensive and integrated understanding of VSMC metabolic dysregulation in the context of vascular diseases.

  1. I recommend that the authors include a perspectives section. I understand that this is found in the conclusions section. However, the latter section is quite long, so it should be shortened.

Response:

Thank you for your valuable suggestion. We have thoroughly revised Section 6 “Conclusion and Future Perspectives” to enhance clarity and coherence.

We have streamlined the section by removing redundant content, ensuring that it is more concise and focused. The core conclusions regarding VSMC metabolic dysregulation in AAD are now clearly summarized, providing a succinct overview of the key findings presented in the review. Additionally, we have separated the discussion of future perspectives into a distinct subsection. This includes a focused exploration of potential pharmacological targets, biomarkers for early diagnosis, and the prospects for personalized therapy. By clearly delineating these elements, we aim to provide readers with a structured understanding of both the current state of knowledge and the avenues for future research. The revised version retains all key insights while improving readability and logical coherence. We believe this revision effectively addresses your concern and presents the review’s conclusions and future outlook in a more organized and accessible manner.

  1. I recommend that the authors review the instructions for citing references again. Please follow the format established by MDPI Publishers. Several references do not follow the required format.

Response:

Thank you for this critical reminder. We have carefully reviewed MDPI’s reference formatting guidelines and comprehensively revised all non-compliant references in the manuscript. We ensured uniform formatting of authors, titles, journals, volumes, and page numbers to fully meet the publisher’s requirements.

Round 2

Reviewer 1 Report

Comments and Suggestions for Authors

Authors have added  a lot of work in the revised version which fully addressed my questions and suggestions.

Author Response

Comments and Suggestions for Authors:

Authors have added  a lot of work in the revised version which fully addressed my questions and suggestions.

Response:

We sincerely appreciate your positive feedback and invaluable guidance throughout the revision of our manuscript. Your recognition that our supplementary work has fully addressed your prior questions and suggestions is a great encouragement to our team.We are grateful for your constructive input that helped elevate our work and kindly request your final consideration for publication.

Reviewer 2 Report

Comments and Suggestions for Authors

The authors have made substantial improvements to the manuscript. The integration of mechanistic pathways and the clinical perspective is clearly enhanced in the later sections.

However, despite revisions to Sections 1–3, these early parts still contain material that is somewhat redundant with previously published and well-cited review articles. Much of the introductory content repeats background information that is already well established in the field, and much of it is not well connected to the later sections.

To improve clarity and reduce redundancy, the following refinements are recommended:

1. Select more carefully the content included in Sections 1–3 where prior literature has already provided detailed coverage.

2. Clarify how each mechanism in the earlier sections contributes specifically to pathways discussed in rest of parts with the new set of figures.

These changes would help sharpen the manuscript’s focus and enhance its originality and value for the journal’s readers.

Author Response

Comments and Suggestions for Authors:

The authors have made substantial improvements to the manuscript. The integration of mechanistic pathways and the clinical perspective is clearly enhanced in the later sections.

However, despite revisions to Sections 1–3, these early parts still contain material that is somewhat redundant with previously published and well-cited review articles. Much of the introductory content repeats background information that is already well established in the field, and much of it is not well connected to the later sections.

To improve clarity and reduce redundancy, the following refinements are recommended:

  1. Select more carefully the content included in Sections 1–3 where prior literature has already provided detailed coverage.
  2. Clarify how each mechanism in the earlier sections contributes specifically to pathways discussed in rest of parts with the new set of figures.

These changes would help sharpen the manuscript’s focus and enhance its originality and value for the journal’s readers.

Response:

We sincerely appreciate the reviewer's constructive feedback. In response, we have significantly streamlined Sections 1–3 to reduce redundancy and eliminate details of well-established background information that is thoroughly covered in existing literature. Notably, we have condensed or removed broad epidemiological data, extensive clinical descriptions, and textbook-level definitions of disease entities, retaining only the essential context to frame our discussion around VSMC dysfunction and metabolic dysregulation. In parallel, we strengthened the internal logic by adding clearer “bridging” statements that explicitly connect each early mechanism (phenotypic switching, cell death, ECM remodeling, and epigenetic regulation) to the downstream metabolic/mitochondrial pathways emphasized in later sections, consistent with the new figure set and the integrative “network map” concept of this review.

Reviewer 3 Report

Comments and Suggestions for Authors

I have re-evaluated the revised version of the manuscript by Shi et al. I thank the authors for their detailed responses to each of my six questions. The manuscript is now considerably improved and has greater scientific rigor.

The expanded objectives of the manuscript are now more specific and consistent with the overall content of the review.

The figures clearly illustrate the importance of the processes described. Furthermore, the perspectives and conclusions section is now more comprehensive.

Author Response

Comments and Suggestions for Authors:

I have re-evaluated the revised version of the manuscript by Shi et al. I thank the authors for their detailed responses to each of my six questions. The manuscript is now considerably improved and has greater scientific rigor.The expanded objectives of the manuscript are now more specific and consistent with the overall content of the review.The figures clearly illustrate the importance of the processes described. Furthermore, the perspectives and conclusions section is now more comprehensive.

Response:

We sincerely appreciate your positive feedback and invaluable guidance throughout the revision of our manuscript.We are grateful that our expanded research objectives are now more specific and better aligned with the overall content of the review. We also value your affirmation that our revised perspectives and conclusions section has achieved greater comprehensiveness, which has further strengthened the academic value of this review. Your constructive input has been instrumental in elevating the quality of our work, and we kindly request your final consideration for the publication of our manuscript.